# Microcantilever-integrated photonic circuits for broadband laser beam scanning

Saeed Sharif Azadeh [1] ✉, Jason C. C. Mak[2], Hong Chen[1], Xianshu Luo [3], Fu-Der Chen[1,2], Hongyao Chua[3], Frank Weiss[1], Christopher Alexiev[2], Andrei Stalmashonak[1], Youngho Jung[1], John N. Straguzzi[1], Guo-Qiang Lo[3], Wesley D. Sacher[1] & Joyce K. S. Poon [1,2] ✉

Laser beam scanning is central to many applications, including displays, microscopy, three-dimensional mapping, and quantum information. Reducing the scanners to microchip form factors has spurred the development of very-large-scale photonic integrated circuits of optical phased arrays and focal plane switched arrays. An outstanding challenge remains to simultaneously achieve a compact footprint, broad wavelength operation, and low power consumption. Here, we introduce a laser beam scanner that meets these requirements. Using microcantilevers embedded with silicon nitride nano-photonic circuitry, we demonstrate broadband, one- and two-dimensional steering of light with wavelengths from 410 nm to 700 nm. The micro-cantilevers have ultracompact ~0.1 mm$^2$ areas, consume ~31 to 46 mW of power, are simple to control, and emit a single light beam. The micro-cantilevers are monolithically integrated in an active photonic platform on 200-mm silicon wafers. The microcantilever-integrated photonic circuits miniaturize and simplify light projectors to enable versatile, power-efficient, and broadband laser scanner microchips.

Optical beam steering is important for engineered light projection in numerous applications including displays[1,2], microscopy[3–6] light detection and ranging (LiDAR)[7,8], communications[9,10], and ion/atom manipulation in quantum processors[11–13] Most commonly, beam scanning is implemented with discrete components, such as galvo-scanners, micro-electromechanical systems (MEMS) mirrors, or acousto-optic deflectors[14]. In recent years, the demand to reduce the size of beam scanners into microchips for easier integration into products has motivated rapid advances in optical phased arrays (OPAs) and focal plane switch arrays (FPSAs) using silicon (Si) photonic integrated circuit (PIC) technology[15–17] FPSAs illuminate discrete points, while OPAs offer continuous angular coverage. Not only can PIC beam scanners minimize size and power consumption, their co-integration with other components, such as photodetectors and lasers, onto a single chip can substantially simplify packaging and reduce

costs compared to assemblies of light sources and beam deflectors[15,18–27], OPAs and FPSAs are very large-scale PICs consisting of arrays of hundreds to tens of thousands of grating coupler light emitters; and to date, they have predominantly been demonstrated in the infrared (IR) spectral region. To achieve two-dimensional (2D) beam steering in OPAs, wavelength sweep and phase-shifters are typically used in conjunction to reduce PIC complexity[23–27] In the visible spectrum, OPAs and FPSAs are even more challenging to realize due to the lack of compact wavelength-tunable lasers and the lower efficiency of phase shifters and switches. Furthermore, the wavelength degree of freedom cannot be used for applications that require specific wavelengths, such as displays and the excitation of atomic transitions. Another major obstacle is that the half-wavelength pitch criterion for single-lobe emission in an OPA[15] is hard to satisfy in the visible spectrum without introducing significant inter-waveguide

[1]Max Planck Institute of Microstructure Physics, Weinberg 2, 06120 Halle, Germany. [2]University of Toronto, Department of Electrical and Computer Engineering, 10 King's College RoadON M5S 3G4 Toronto, Canada. [3]Advanced Micro Foundry Pte. Ltd., 11 Science Park Road, Singapore Science Park II, Singapore 117685, Singapore. ✉e-mail: sazadeh@mpi-halle.mpg.de; joyce.poon@mpi-halle.mpg.de

crosstalk or reducing the minimum feature size. Recent demonstrations of visible-light beam scanners have been based on OPAs[28–33] using an external super-continuum source coupled with a tunable filter[33] or a rotation stage to scan the beam in the second dimension while requiring high on-chip driving powers of 2 W[31]. These approaches are difficult to miniaturize into single chips in the foreseeable future.

Here, for the first time to our knowledge, we demonstrate visible spectrum 1D and 2D beam scanner PICs that emit single beams without any sidelobes and at arbitrary wavelengths. The scanners consist of MEMS cantilevers with integrated silicon nitride (SiN) nanophotonic waveguides and grating couplers. The electrical drive power to cover the full scanning range of each axis was <31 mW, about 2 orders of magnitude lower than previously reported visible light PIC beam scanners[31]. When resonantly driven, a scanning rate in the range of 10–100 kHz was achieved dependent on the cantilever length. In contrast to the recent Si waveguide MEMS phase-shifters and beam scanners[16,31,34], our approach does not require waveguiding in Si or electrically conductive Si. Our cantilevers are agnostic to the waveguide core material; hence, they apply to SiN waveguides, which are optically transparent at visible wavelengths. Using standard fabrication processes in Si photonics foundries, our cantilever devices were monolithically integrated within a foundry-manufactured visible spectrum PIC platform on 200-mm Si (Fig. 1a, b), in which other components including low-loss wideband edge couplers, high quantum efficiency waveguide photodetectors, low crosstalk junctions, and efficient thermo-optic phase shifters have been reported[35–38]. Due to the mechanical nature of the actuation, identical steering ranges were achieved for wavelengths between 410 and 700 nm. Because our approach does not require a large array of light emitters, excluding the laser source, our PICs possess the smallest footprint amongst all chip-scale beam scanners to date of 0.14 mm × 1.1 mm for 2D scanning. These versatile MEMS cantilevers can also be easily implemented in generic silicon-on-insulator (SOI) photonic platforms with heaters, deep trenches, and an undercut step, opening new directions for compact beam-scanning PICs.

## Results

### Operation principle and architecture

We realized two types of beam scanner designs: (1) a rectilinear MEMS cantilever (Fig. 1c, e), which steered the output beam only in the longitudinal direction, and (2) an L-shaped singly clamped cantilever (Fig. 1d, f) capable of beam steering in both the longitudinal and transverse directions via two control voltages. In both cases, light was guided in a SiN waveguide embedded in the cantilever and terminated with an output grating coupler at the distal end. The grating coupler was 10 μm wide and 25 μm long, consisting of fully etched 150 nm thick SiN teeth with a period of 440 nm. It had an average loss of 5.2 dB at wavelengths between 410 and 700 nm (see Supplementary Section 1 for details on the grating couplers). The simulated grating coupler emission full-width at half-maximum (FWHM) beam widths were 0.78° longitudinally and 2.2° transversely, and the measured FWHM widths were 1.4° and 3.1°, respectively. Figure 1a shows the PIC platform cross-section with the cantilever delineated. The suspended structure was formed by a deep trench $SiO_2$ etch followed by an isotropic Si undercut. To actuate the cantilever, we created an electro-thermal bimorph using a 2 μm-thick aluminum (Al) layer atop a 2.5 μm thick $SiO_2$ layer (see Supplementary Section 2 for the layer thickness design). Embedded resistive titanium nitride (TiN) strips heated the cantilever with applied voltages as shown in the circuits in Supplementary Section 3, Fig. S3. Due to the greater thermal expansion of Al compared to $SiO_2$, the cantilever bends downward with increasing temperature.

Figure 1c illustrates the schematic of the rectilinear cantilever for 1D scanning. The width of the cantilever tapered from 30 μm at the proximal end to 15 μm at the distal end. This shape strikes a balance between robustness and thermal efficiency. A wider cantilever base is less likely to crack and has better performance under high stress, while a narrower cantilever width reduces the volume to be heated, and thus higher temperatures could be reached for the same applied electrical power, $P_e$. We designed rectilinear cantilevers with four different lengths ($L_{can}$) of 300, 500, 800, and 1000 μm, with resistances of the TiN ($R_{TiN}$) heaters 430, 480, 550, and 680 Ω respectively. The choice of cantilever length is a trade-off between the steering range and actuation time constant. As will be shown, while the longest cantilever

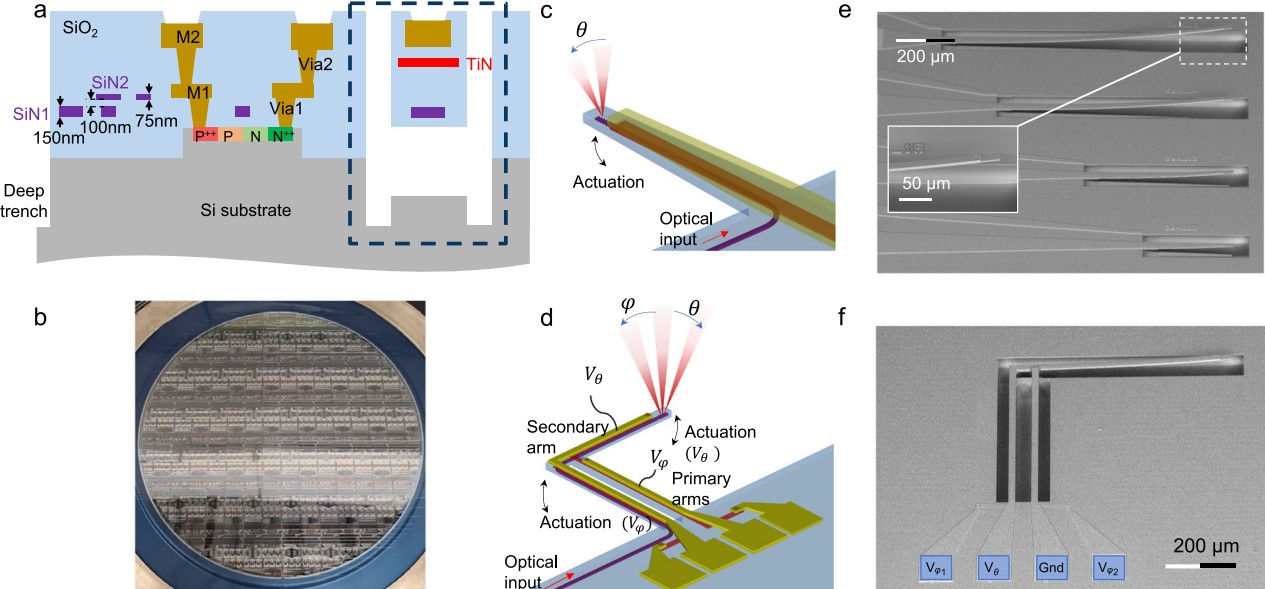

**Fig. 1 | Overview of the microcantilevers. a** Cross-sectional schematic of the integrated photonic platform containing the microcantilevers. The cantilever cross-section is delineated in the dashed box. **b** Photograph of the fabricated 200 mm diameter wafer. **c** Schematic of the rectilinear cantilever for 1D beam scanning. A SiN waveguide with an output grating coupler and a TiN heater are embedded in the cantilever. **d** Schematic of the L-shaped cantilever for 2D beam scanning in the longitudinal ($\theta$) and transverse ($\varphi$) directions. Scanning electron micrographs (SEMs) of **e** rectilinear cantilevers (with 300, 500, 800, and 1000 μm lengths) and **f** an L-shaped cantilever. Due to film stress, the cantilevers bend upwards in the absence of applied electrical power.

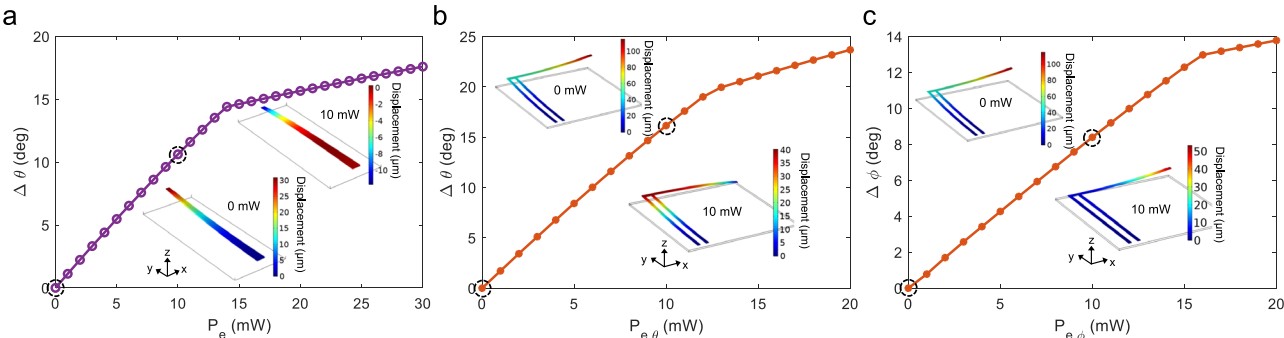

**Fig. 2 | Simulated beam steering of the rectilinear and L-shaped micro-cantilevers. a** Simulated angular scan range of the 500 μm long rectilinear cantilever vs. applied electrical power ($P_e$) to the TiN layer; insets show the simulated shape of the cantilever under 0 mW (bottom) and 10 mW (top) of applied power. **b** Simulated angular scan ranges of the L-shaped cantilever along the $\theta$-axis and

$\varphi$-axis. The lengths of the primary and secondary arms are 500 and 600 μm, respectively. The insets show the calculated displacement of the L-shaped cantilever under 10 mW (bottom) and 0 mW (top) electrical power applied to the **b** secondary arm and **c** primary arm. In the plots, the markers are the simulation results, and the dashed circles indicate the operating points of the insets.

achieved the largest steering range, the shortest devices were faster due to a lower thermal time constant.

In the absence of applied electrical power, $P_e = 0$ mW, the cantilever bent upwards due to the initial stress between the metal and oxide, which were deposited at different temperatures. A scanning electron micrograph (SEM) of the rectilinear cantilevers captured at a 45° tilt shows this expected initial upwards bending (Fig. 1e). The simulated angular steering range as a function of $P_e$ at $L_{can} = 500$ μm is shown in Fig. 2a. The top and bottom insets of Fig. 2a illustrate the downward displacement of the cantilever tip under applied power. The tuning efficiency ($d\Delta\theta/dP_e$) is reduced at $P_e > 14$ mW as the cantilever tip contacts the Si substrate at the bottom of the undercut, modeled to be 25 μm deep, yet continues to bow (see Supplementary Section 3 for details). The initial strain was calculated assuming the deposition temperature of Al to be 88 °C to match the experiment. The simulations predict an angular scan range of 17.6° and 29.5° respectively for the 500 and 1000 μm long cantilevers at $P_e = 30$ mW.

Figure 1d shows a schematic of the L-shaped cantilever for 2D beam scanning. The design has two primary arms that tilt the grating coupler in the transverse direction under a control voltage $V_\varphi$ and a secondary arm which tilts the beam in the longitudinal direction with a control voltage $V_\theta$. An SEM image of the device is shown in Fig. 1f. The primary and secondary arms were 500 and 600 μm long, respectively, with a constant width of 20 μm. The SiN waveguide in the cantilever had a 40 μm bend radius to connect the input to the grating coupler at the cantilever tip. Figure 2b and c show the simulated beam steering along the $\theta$-axis with respect to applied power to the secondary arm ($P_{e,\theta}$), and along $\varphi$-axis with respect to electrical power applied to the primary arms ($P_{e,\varphi}$), respectively. Thermal crosstalk between the primary and secondary arms causes a slight $\theta$-axis tilt under $P_{e,\varphi}$, and vice versa (see Supplementary Section 3), but is sufficiently small to allow for independent control of the beam direction along the two angular axes. Again, the reduction in the angular tuning efficiency is due to the tip of the cantilever contacting the substrate. The simulations predict a maximum beam steering of 23.7° along the longitudinal direction under $P_{e,\theta} = 20$ mW, and 13.8° under $P_{e,\varphi} = 20$ mW.

## Measurements

The far-field radiation pattern was captured using the setup described in the Methods Section and illustrated in Fig. 3a, where the input laser light is coupled from a multi-wavelength laser source via a single-mode fiber onto the chip. Unless otherwise stated, the input polarization was set to transverse-electric (TE) mode to maximize the optical transmission of the grating coupler. The recorded far-field image of the grating output shows a divergence angle of 1.4° in the longitudinal direction ($\theta$) and 3.1° in the transverse direction ($\varphi$) at $\lambda = 488$ nm. As

the beam steering range is independent of the laser wavelength due to the mechanical nature of the actuation, $\lambda = 488$ nm was chosen as a representative wavelength for measurements unless otherwise stated. Figure 3b visualizes the extent of the steering range of the four rectilinear cantilevers, by capturing an overlay of the far-field images at applied powers of 0 mW (right beams) and 20 mW (left beams).

The measured beam steering angles as a function of the applied electrical power are shown in Fig. 3c. We measured maximum beam scanning ranges of 11°, 17.6°, 22.6°, and 30.1° with 30 mW applied electrical power, respectively, for 300, 500, 800, and 1000 μm long cantilevers, in good agreement with the simulated values (dashed lines). This corresponds to about 8, 12, 16, and 21 resolvable points at $\lambda = 488$ nm, respectively for the shortest to the longest cantilevers. The power efficiency in terms of $\frac{d\Delta\theta}{dP_e}$ of the 1000 μm cantilever was 1°/mW and was lower for shorter cantilevers due to the heat sinking effect of the metal contacts, which reduced the effective temperature in the proximal end of the cantilever. As predicted by the simulations and attributed to the cantilever coming into contact with the substrate (Fig. 2, Supplementary Section 3), a reduction in the angular tuning efficiency was observed.

The non-resonant scan rate was mainly limited by their thermal time constant. We measured the time response of the devices by applying a 10 mW pulsed signal with various duty cycles and recording the far-field pattern (see "Methods" section and Supplementary Section 4). Figures 3d, e, respectively, show the measured 10%-to-90% fall and rise time responses of the 300 μm long cantilever (see Figures S4 and S5 for the time response of the other beam scanners). We measured an average response time of 1.2, 2.6, 4.1, and 4.7 ms, respectively, for the shortest to the longest cantilevers. These response times are comparable to the fastest liquid crystal switches[39]. However, many beam steering applications (including displays, 3D sensing, and microscopy) require only periodic scanning of light beams in at least one direction and not aperiodic switching. Thus, to reach higher scanning rates, we drove the cantilevers at their resonance frequencies, beyond the electro-thermal time-constant limit (Fig. 3f). The resonance frequencies of the devices were 5.7, 11.8, 24.8, and 77.4 kHz respectively for 1000, 800, 500, and 300 μm long cantilevers. At these frequencies, maximum beam scan ranges of 12°, 10.4°, 11.3°, and 9.8° were achieved under 20 mW of applied electrical power, respectively.

The measured angular scan ranges of the L-shaped cantilever are shown in Fig. 4a, b, respectively, in the longitudinal and transverse directions. The lengths of the primary and secondary arms were 500 and 600 μm, respectively. An angular steering range of 24.0° and 12.2° was achieved with a maximum applied electrical power of 23 mW, respectively, for the $\theta$-axis and $\varphi$-axis. To illustrate the 2D angular range of the steering in Fourier space, we applied two different signals

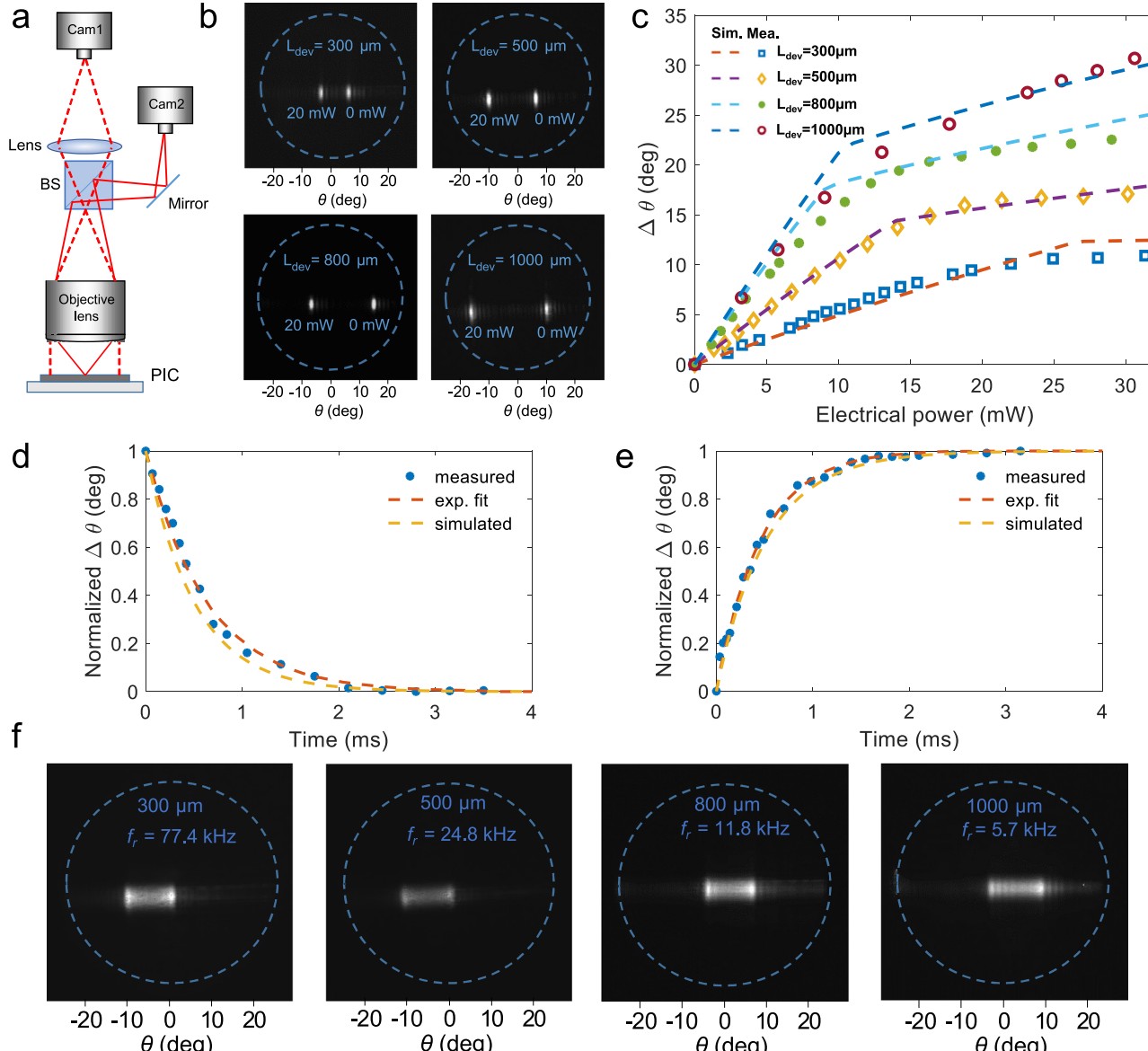

**Fig. 3 | Rectilinear cantilever: Experimental setup, far-field patterns, DC, and time-dependent response. a** Schematic of the imaging setup using a regular lens for creating a Fourier image, and a beam splitter (BS) for the simultaneous capture of the near- and far-field patterns. The solid lines and dashed lines show, respectively, the near-field and far-field trajectories imaged on the cameras Cam 2 and Cam 1. **b** Measured far-field patterns of the grating output at 0 mW (right beams) and 20 mW (left beams) for the four cantilever lengths. The two outputs are overlaid on the same image by applying a 2 Hz step function and capturing the grating far-field output over a 1-second exposure. For $L_{dev} = 1000\,\mu m$, the measurement setup was slightly shifted to capture both spots. **c** Measured (markers) and simulated (dashed lines) beam steering versus applied DC electrical power to the rectilinear cantilevers. **d** Fall time and **e** rise time of the 300 μm long cantilevers. **f** Far-field images of the rectilinear cantilevers captured at their respective resonance frequencies.

to each set of arms. A 120 Hz sinusoidal electrical signal with a peak-to-peak voltage of 4 V and a DC offset of 2 V was applied to the secondary arm, while a 30 Hz electrical signal with a peak-to-peak voltage of 3 V and DC offset of 1.5 V was applied to the primary arms. The corresponding far-field image of the output beam captured over a 33 ms exposure time is shown in Fig. 4c, covering a range of ~ 24° × 12° in Fourier space.

The 10–90% rise time of the primary and the secondary arms were 4.3 and 4.7 ms, but faster beam steering is possible on resonance. Figure 4d shows the simulated first (left) and second (right) resonance modes of the L-shaped cantilevers, which were at 6.9 kHz and 14.3 kHz, respectively. At the first resonance, both the secondary and the primary arms were in phase, simultaneously moving upward (or downward) thus moving the Fourier image of the far-field beam in the negative (or positive) directions of the θ- and φ-axes. Experimentally,

the first resonance frequency was found to be 7.6 kHz, with its far-field radiation pattern shown in Fig. 4e. At the second resonance frequency, measured to be 17.4 kHz, the displacement of the primary and secondary arms had a π-phase difference, resulting in the far-field pattern in Fig. 4f. To excite the resonances, a pulsed voltage with an average power of 10 mW was applied to the primary arms of the L-shaped cantilever. These resonances can be excited in linear superposition, as shown in Fig. 4g.

As a proof-of-concept demonstration, we used the cantilever to project a 2D image of our department name "NINT" (Fig. 4h). The applied voltages to the primary and secondary arms were controlled by a computer to steer the beam in the Fourier space in the desired directions. The images were generated without controlling the laser output power, and the pattern formation relied solely on the actuation of the beam. The refresh rate of the image was set to 30 Hz over a 33 ms

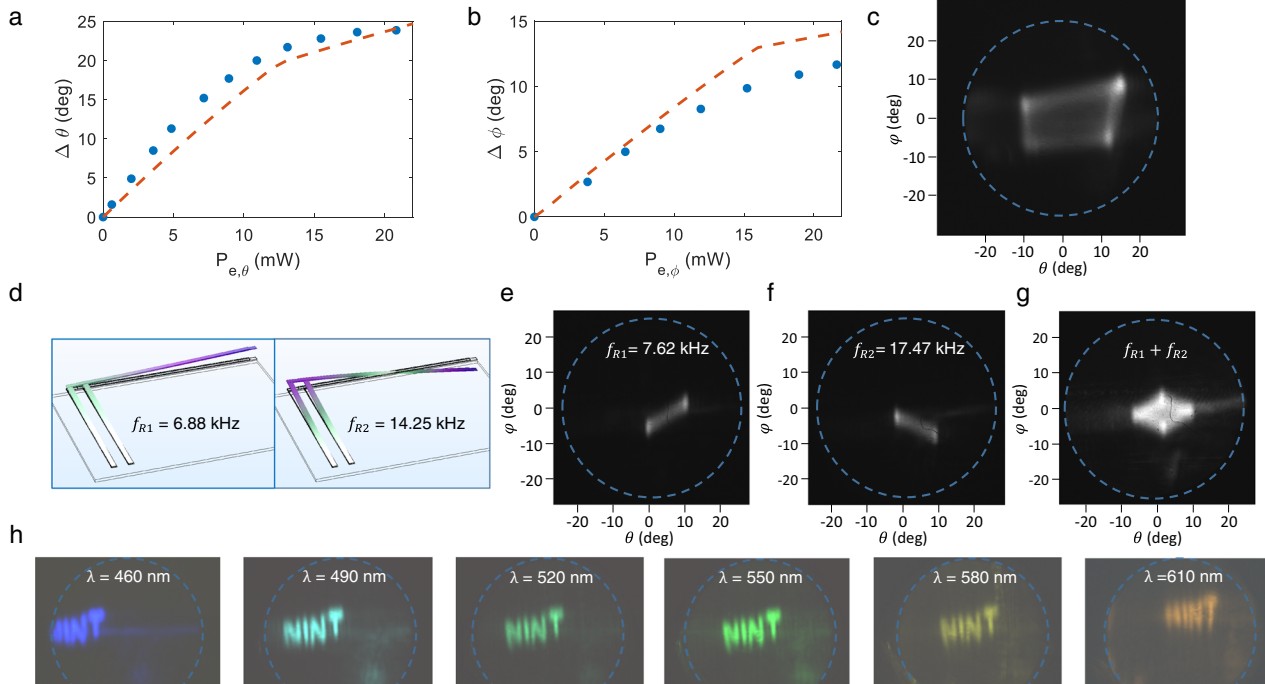

**Fig. 4 | L-shaped cantilever characterization. a** Measured steering of the output beam in the longitudinal (transverse) direction vs. the electrical power applied to the secondary (primary) arm (**b**). **c** Recorded far field pattern of the L-shaped cantilever output under a drive voltage $V_\theta = 2[\sin(2\pi f_1 t)+1]$ V, where $f_1 = 120$ Hz, applied to the secondary arm, and $V_\phi = 1.5[\sin(2\pi f_2 t)+1]$ V, where $f_2 = 30$ Hz applied to the primary arms. **d** Simulated resonance modes of the L-shaped cantilevers at the first and second resonance frequencies. Far-field images of the device are recorded at the first (**e**) and the second (**f**) resonance frequencies, and superposition (**g**) of the two resonances. **h** Images produced by the L-shaped cantilever scanning our department name ("NINT"), demonstrating the potential for image projection.

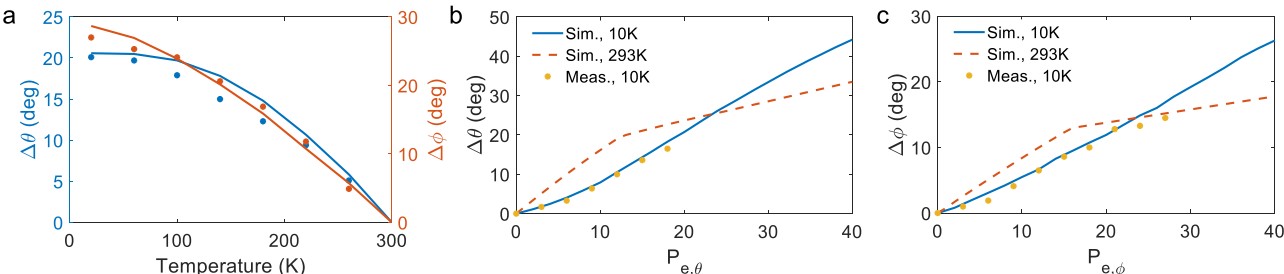

**Fig. 5 | Cryogenic characterization of the L-shaped microcantilever. a** Simulated (solid lines) and measured (dots) values of the initial angle shift vs. the ambient temperature in the longitudinal (left axis) and transversal (right axis) directions, without applying electrical power. **b** Simulated and measured steering of the output beam in longitudinal (transverse) directions (**c**).

exposure time. The experimental results of 1D and 2D beam scanners are summarized in Table S1.

Finally, we characterized our beam scanner in cryogenic conditions for several reasons: (1) the low temperature increases the intrinsic stress in the cantilever and poses a higher risk for device failure, (2) electrothermal devices may not work in a cryostat since their thermal budget can overwhelm the cooling power of the cryostat, and (3) some applications, for example in quantum control, may prefer to have the PIC inside the cryostat[11,12]. We tested another L-shaped cantilever device in a cryostat at temperatures as low as 10 K (see "Methods" section and Fig. S8 for details on the cryogenic unit). By reducing the temperature, the cantilever was further bent upwards due to the higher expansion coefficient of Al compared to $SiO_2$. Therefore, the initial angle of the output beam was increased in both $\theta$ and $\varphi$ directions, as shown in Fig. 5a, where the simulated initial angle (lines) and the measured values of the tilt (dots) are plotted. The simulated scan range of the L-shaped cantilever versus the applied electrical power in

$\theta$ and $\varphi$ directions are respectively shown in Fig. 5b, c. The dashed line shows the simulated scan range at room temperature for comparison. Due to the higher initial deflections at 10 K compared to room temperature, the beam can deflect a larger angle without contacting the Si substrate; thus, no change in the angular tuning efficiency was observed at up to 40 mW of applied electrical power. The experimental results (yellow dots in Fig. 5b, c) are in good agreement with the simulated tilt of the cantilever. Due to the limited field-of-view of the cryogenic setup (i.e., the emitted beam incident on the sides of the cryogenic chamber), the experimental observations were limited to angles <20°. The cantilever was driven at a resonance frequency of 16.76 kHz for ~7 million cycles at 10 K with no observable degradation. Thermal simulations show that the heat is localized to the cantilever device (Supplementary Section 6). The calculated half-decay length of the temperature, defined as the distance where the difference between the local temperature and the cryostat setting is 50% of the maximum, is 20 μm for 26 mW of applied power, and the temperature on the PIC

drops to below 11 K within 600 μm distance from the edge of the cantilever.

## Discussion

These reported ultracompact, power efficient, monolithically integrated microcantilevers are, to the best of our knowledge, the first side-lobe-free 2D beam scanners for the visible spectrum and the beam scanners with the widest operation wavelength bandwidth spanning >300 THz. Our devices achieved a 2D scan range of 24° × 12° with arm lengths of ~500 μm with only 46 mW of drive power. The beam scanning was achieved without wavelength-tuning or using phase shifters at scan rates of tens of kHz on resonance. The scanning range and power emission of the devices were unchanged after one billion scan cycles on resonance in ambient conditions.

In comparison with the prior art, the microcantilevers offer unique advantages (see Supplementary Section 7 for a comparison chart). Our approach is distinct from the photonic-MEMS phase-shifters using Si waveguides[16,34], since we do not require highly doped Si. It also achieves single-lobe output beam steering without wavelength tuning, which has not yet been possible with visible spectrum OPAs[28,31,40]. The power consumption of the cantilever beam scanners was ~2 orders of magnitude lower than visible-wavelength OPAs, which required ~2 W[31]. Our beam scanners are also distinct from previous MEMS-tunable grating couplers for spectral tuning and fiber-to-chip coupling in the infrared[41–45] First, we have achieved a larger 1D scan range (30° in 1D vs. 5.6° in ref. [44]) and 2D scanning. Second, our design displaces the entire grating emitter rather than tuning the grating period and apodization, so the emission profile minimally deteriorates during cantilever actuation. For example, in ref. [44], due to the wide angular beam width, the number of resolvable points is only about 0.62. In comparison, the number of resolvable spots of the L-shaped cantilever here is about 66, limited by the divergence angle of the grating emission (1.4° in $\theta$-direction and 3.1° in $\varphi$-direction). Lastly, the microcantilevers dramatically reduce the complexity of the drive circuitry for 2D beam scanning compared to large-scale PIC approaches of OPAs and FPSAs – only a single device needs to be controlled with 2 drive signals (for 2D beam scanning) and resonant scanning enables ~100 kHz scan rates. This simplification will reduce the power consumption of the drivers, calibration, control, and stabilization of a large number of array elements.

In comparison to discrete MEMS scanning mirrors co-packaged with a laser diode, the advantages of the monolithically integrated MEMS PIC approach are (1) flexibility to co-integrate PICs to achieve passive and active functionalities, such as wavelength multiplexing, filtering, modulation, and detection, into a single chip without the need to assemble micro-components (e.g., dichroic beamsplitters, lenses); (2) reduced packaging complexity, especially when the lasers are integrated on the chip (a recent demonstration in the near infrared range near 980 nm is reported in[46], and we are also developing heterogeneous laser integration approaches for our platform); (3) compatibility with wafer-scale manufacturing for the MEMS, PIC, and their co-integration, and (4) generally a higher resonance frequency due to the smaller volume of the cantilever compared to a mirror (smaller mirrors are possible but require tighter alignment tolerances). On the other hand, in general, MEMS mirrors have a higher radiation efficiency compared to grating couplers (typically ~90% vs. ~50%, respectively) unless extra features, such as back reflectors, are incorporated on chip, and mirrors have broadband reflection. Thus, the choice between using a co-packaged MEMS mirror and a fully integrated solution depends on the specific application.

The cantilever beam scanners can be extended in several ways. The achieved scan range was limited by the substrate, so it can be increased by making the undercut deeper. Simulations show that with a 120-μm-deep undercut, a 1-mm-long rectilinear cantilever would achieve a deflection of 52° with 30 mW of applied electrical power.

Since the initial deflections are determined by the beam lengths, multiple beam scanners can be used together to expand the scan range. To increase the number of resolvable spots, the divergence angle of the output beam along the propagation direction could be significantly decreased by >10× using weaker and longer gratings (Supplementary Section 5), and the divergence in the lateral direction can be reduced with an array of cantilevers that effectively forms an OPA. To reduce power consumption, electro-static or piezoelectric actuators can be incorporated instead of the electro-thermal bimorph design[47]. Beyond the demonstrated geometry, other planar-compliant mechanisms, as well as other types of light emitters, such as edge couplers (akin to fiber scanners), sub-wavelength waveguides, OPAs, and metasurfaces, can be used. The deformation along the cantilever may also be exploited as a tuning method for or a sensor using embedded nanophotonic devices.

The reliability of the cantilevers and other MEMS structures implemented in the PIC platform can be systematically characterized in the future in more mature and packaged devices as a function of temperature, humidity levels, mechanical shocks, applied electrical power, and the scan range and speed[48]. In the ideal operation scenario, the microcantilever tip should not come into contact with the substrate, which can be achieved in the future with a deeper undercut. Due to the difference in the coefficients of thermal expansion of $SiO_2$ and Al, the cryogenic conditions in our experiments (at 10 K) increase the intrinsic stress in the cantilevers compared to an elevated temperature (in simulation, the stress at clamping is ~10 times higher at 10 K compared to the room temperature). Although not a comprehensive reliability characterization, our device measurements at 10 K for 20 minutes and in the ambient for 3 days did not exhibit any observable degradation; these observations are a promising indication of the robustness of the microcantilever-integrated photonic circuits.

In summary, the microcantilever-integrated photonic circuits demonstrated here open exciting avenues for photonic beamforming. The approach decouples the design of the scan range from the light emitter. The cantilevers are simple to control, can be incorporated in any photonics platform possessing an undercut etch, and can be placed anywhere within a chip. Microcantilever-integrated photonic integrated circuits may enable ultracompact and power-efficient solutions to transform augmented reality displays, microscopy, quantum information processors, and 3D mapping technologies.

## Methods

### Numerical simulations

Electro-thermomechanical simulations of the MEMS structures were performed using finite element method (FEM) in COMSOL Multiphysics to find the resonance modes of the cantilevers, as well as their displacements with respect to the applied voltage, and their time response. The Young's modulus of SiN, TiN, Al, and $SiO_2$ were assumed to be 250, 500, 70, and 73 GPa and their Poisson ratio was set to 0.23, 0.25, 0.33, and 0.17, respectively. For thermal simulations, the Si substrate and electrical pads were set to a constant temperature of 293 K. The thermal expansion coefficients of Al and $SiO_2$ were set to $23 \times 10^{-6}$ and $5.5 \times 10^{-7}$ 1/K, respectively, with thermal conductivities of 238 and 1.4 W/mK. Optical simulations of the grating couplers were carried out using the 3D finite difference time domain (FDTD) method in Lumerical software. The refractive indices of SiN and $SiO_2$ were assumed to be 1.81 and 1.46 at $\lambda = 532$ nm.

### Device fabrication

The devices were fabricated on 200-mm diameter Si wafers at Advanced Micro Foundry (AMF) as part of our visible-light photonic integrated circuit platform. The fabrication process included steps to implement other devices in this platform. It started with ion implantation and partial etching of the Si substrate to define the

photodetectors[36]. Next, a $SiO_2$ layer as the bottom cladding of the waveguides was formed using PECVD. Then a SiN layer with the targeted thickness of 150 nm was deposited atop the oxide layer in a PECVD process. The SiN waveguides were then defined by 193 nm deep ultraviolet (DUV) lithography followed by a reactive ion etching (RIE) step. Additional $SiO_2$ and SiN deposition and patterning steps were performed to define a second 75 nm thick SiN waveguide layer to form low-loss bi-layer edge couplers[35]. The layers were planarized using chemical mechanical polishing. Next, a TiN layer was deposited and patterned to be used as a heater, followed by two Al layers and oxide openings for bond pads. The top Al layer thickness was 2 μm to enhance the strain and thus the initial displacement of the cantilevers. Finally, to suspend the MEMS structures and to form the $SiO_2$ bridges in our thermal phase-shifters[38], a deep trench was formed followed by undercut etching of the Si.

### Room temperature measurement setup

Figure 2a shows the experimental setup for device characterization. The setup captured the emission pattern in real and Fourier space imaging modes. The far-field output was collected by a high numerical aperture objective lens (with 20× magnification, NA = 0.42, and effective focal length = 10 mm) to project the far-field radiation pattern into the Fourier plane, where it was captured by a CMOS camera. We utilized an uncollimated white light source to visualize the sample surface (not shown in Fig. 2a). For simultaneous visualization of the near-field and facilitating the alignment procedure, a beam splitter diverted half of the radiated beam to a second CMOS camera. Light from a multiwavelength laser source (Coherent OBIS Galaxy) was edge-coupled to the chip through a single-mode fiber (Nufern S405-XP) with an inline polarization controller. The polarization was set to transverse electric (TE) mode to maximize the optical transmission of the grating coupler.

### Time response measurements

To measure the temporal response of the cantilevers, we coupled light into each device and recorded the far-field radiation under an applied periodic electrical pulse with a peak power of 10 mW and varying duty cycles. In the case of a 50% duty cycle, provided that the period of the square pulse ($T$) was much longer than the rise/fall time of the cantilever ($t_r$), the maximum displacement of the far-field beam would be equal to the results for a DC voltage (the first far-field image in Fig. S4a). By reducing the duty cycle to a level below the rise time of the device, the emitted beam trajectory became shorter, allowing us to determine the transient response of the device. Measurements of the far-field trajectories are shown in Supplementary Section 4.

### Cryogenic measurement setup

The cryostat was taken to a vacuum at a pressure of $<10^{-4}$ mbar and the temperature was reduced by liquid Helium (He) cooling. The temperature of the cold head was controlled using a 100 W built-in heater connected to an automatic PID controller, which could also set the He flow using a magnetic valve. To establish electrical connectivity, the PIC was mounted on a custom printed circuit board (PCB) using a thermally conductive epoxy (Loctite 84-1LMIT1) and then wire bonded to a PCB (Fig. S8). The wires as well as the optical fiber were routed inside the chamber via electrical and optical high-vacuum fit-through adapters. The optical fiber was attached on top of the PCB using a transparent optical adhesive (DYMAX OP-4-20632) and cured with UV light while being actively aligned to the input edge coupler.

### Reporting summary

Further information on research design is available in the Nature Portfolio Reporting Summary linked to this article.

## Data availability

Data underlying the results presented in this paper are available at https://doi.org/10.17617/3.AT47OS. Additional data are available from the authors on request.

## Code availability

The code used in this study is available from the authors on request.

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

## Acknowledgements

The authors thank Dr. Holger Meyerheim for lending the cryostat. J.K.S.P. acknowledges support from the Natural Sciences and Engineering Research Council of Canada (grant number: RGPIN-2018-06491). Open Access of this publication is enabled and organized by Projekt DEAL.

## Author contributions

J.C.C.M proposed the initial device concept. S.S.A and J.C.C.M performed electromechanical simulations. The layout was generated by S.S.A. and W.D.S.; X.L., H.C., and G.Q.L. were responsible for device fabrication. A.S. and Y.J. established the measurement setup, with contributions by H.C., F.D.C., and C.A. for the experiments. J.N.C. performed the wirebonding. S.S.A. and H.C. characterized the devices. F.W. and S.S.A. performed the cryogenic measurements. W.D.S., J.K.S.P., H.Chua, and X.L. conceived the photonic platform. S.S.A and J.K.S.P. co-wrote the manuscript with inputs from co-authors. J.K.S.P. supervised and provided feedback on all aspects of the project.

## Funding

## Competing interests

The authors declare no competing interests.
