## [Peer Review File · Nature Communications]

REVIEWER COMMENTS

Reviewer #1 (Remarks to the Author):

The authors report an innovative design for laser beam scanning, which relies on the thermal manipulation of delicate microcantilevers integrated monolithically with a nanophotonic grating coupler circuit. The bending of cantilevers was implemented through inducing thermal expansion over an electro-thermal bimorph with two different materials. Consequently, the output beam deflects in response to the electric power. Two cantilever designs, i.e., rectilinear and L-shaped cantilevers, were prepared for beam scanning in 1D and 2D plane, respectively. For each design, a number of samples with geometrical variations were compared showing a tradeoff between the steering efficiency and actuation time. Further demonstration of image projection was also done by applying electric power in both arms of a L-shaped cantilever. A maximum 2D scanning range of $24^\circ \times 12^\circ$ was achieved with a low driving power ~ 46 mW for an arm length of ~ 500 μm . In comparison to other beam scanning designs, the authors claim its advantages of side-lobe free spectrum, compact footprint, broadband operation and low power consumption.

The work and its conclusions appear well supported by the data. I agree that this is original research towards visible light beam scanning with large field of view and low power consumption. And this work will be of significance to enrich the development of power-efficiency solutions in visible beam steering applications in AR, microscopy and light detection and ranging (LiDAR) etc.

However, the impact and significance may not meet the standard of Nature Communications provided that, a) the FOV seems limited by undercut of silicon substrate;

b) a tradeoff between scanning rate and the steering range, and the scanning rate was limited due to the mechanic nature of the cantilever in the range of only 10-100 kHz;

c) also, the size of cantilever basically lies in the micron-meter scale, which is highly susceptible to the ambient variations such as the temperature change. The operation reliability should be verified.

Regarding to the above, I WOULD NOT recommend to publish this work in Nature Communications unless below comments are addressed:

1)The authors illustrate the circuit of compact footprint, broadband operation and low power consumption. However, I can not see the improvement of such a monolithic design in compared to a combination design of an on-chip laser with an external MEMS reflector. However, the cost and fabrication complexity are distinct.

2)I would recommend the authors to provide a group of data to show the reliability of microcantilevers at different temperature ranges, such as the beam deflection without power input at 20 - 70 °C?

3)The circuit contains a single grating coupler of 10 μm wide and 25 μm long. This single grating element will be fundamentally diverged in far field, and the beam divergence is noticeable in free space, particularly in transverse direction, highly dependent on the aperture size. Without any beam concentration on the emitting element, just like OPA, this will be a big issue for practical application.

Even though the smaller divergence angle with longer grating is claimed by the authors, there will be additional issue appearing together with longer gratings. 0.01° divergence is delivered by 8 mm grating aperture reported in [C. V. Poulton, et al., IEEE J. Sel. Top. Quantum Electron 28(5), 6100508(2022).] The authors should seriously discuss how this millimeter size grating is bended by a much bigger microcantilever. Will the thermal actuation affect the optical properties of millimeter-size grating? How should the thermal isolation be conducted? What scanning rate can be expected on this bigger cantilever?

4)Can the authors explain more about how you assumed the initial strain of cantilevers to match the experimental observation by seeking the deposition temperature of Al in the simulations?

5)Please re-check the caption of Figures 2b and 2c upon the illustration of primary and secondary arms. The illustration should be consistent with Figure 1d.

6)In Figure 3b, I suppose that all 4 samples were prepared under same conditions. Then, the initial strain (right patterns) of GC tips should be of slight increase regrading to the growth of arm length. This can be verified in Figure 3b with $L_{dev} = 300, 500$ and $800 \mu\text{m}$. However, the initial value for $L_{dev}=1000 \mu\text{m}$ seems not greater than others. Can you explain why?

7)The dynamic motion of cantilevers must be distinct from their static response. If possible, please provide the frequency-amplitude(strain) response of the rectilinear cantilever tests at one certain arm length. And also, for $L_{dev}=1000 \mu\text{m}$, please also clarify why the resonance strain is less than static strain, if you compare the results in Fig. 3f and Fig. 3c.

8)Would you please roughly explain the possibility of cantilever damage under large strain resonance, since the cantilever may have chance to knock onto the silicon surface at high frequencies. This test would support the robustness of the microcantilevers.

9)Frequency response of cantilevers under cryogenic conditions should be provided, such as the response time as well as the resonance steering range.

Reviewer #2 (Remarks to the Author):

The authors demonstrated a laser beam scanner by using microcantilevers embedded with silicon nitride nanophotonic circuitry. They achieved broadband, one- and two-dimensional steering of light with a broad visible wavelength range. The device has a compact size and low power consumption. It was realized in an active photonic platform on 200-mm silicon wafers. The design is novel and interesting. Good performances were obtained. I have the following questions and comments.

1. It is not clear how the electrical connections were formed in the rectilinear cantilever and L-shaped cantilever. In order to generate heat by the applied voltage, the two ends of the TiN heater must be connected the aluminum electrodes. Is the connection through the via in the cantilever? In Fig. S3, they

provide the circuit diagrams for the L-shaped cantilever, but it is hard to match the circuit diagrams with the structure diagram.

2. In Fig. 1(a), the cantilever part includes a part of the silicon substrate (the undercut is not just under the SiO₂-Si interface). How can this be realized in the device fabrication?
3. For the L-shaped cantilever, why did they design two primary arms instead of one? What is the lateral separation distance between the two primary arms? How will this distance affect the beam scanning in both longitudinal and transverse directions? When electrical power is applied to the primary arms to get beam steering along the ϕ -axis, is the power evenly applied to the two arms?
4. According to Fig. 1(d), the SiN waveguide is first routed through the left primary arm and then the secondary arm before terminated at the grating coupler. In other words, there is no SiN waveguide in the right primary arm. The two primary arms are not identical. Will this asymmetry make the tuning efficiency different for the two primary arms and hence cause θ -axis tilt under ϕ -axis tuning?
5. In Fig. 4(c), the applied electrical voltage signals are two sinusoidal waves. The voltage expressions in the caption are wrong (the sin function missing).
6. What is the purpose of characterizing the beam scanner in cryogenic conditions? Is there any special application that requires the beam scanner to work in a very low temperature?
7. In Fig. S2(b), the temperature suddenly drops at around 470-micron distance along the cantilever. According to my understanding, this end is not clamped. Then what is the reason  that causes this temperature drop?
8. Fig. S3(d) caption says the graph shows the calculated ϕ -axis angular tilt of the grating coupler. However, the y-axis of the graph is labeled as $\Delta\theta$ (it should be $\Delta\phi$).
9. If the duty-cycle of the applied square pulses is varied between 1% to 50%, the rise time can be measured, but how can they get the fall time? According to my understanding, in the fall time measurement, the duty-cycle should be varied between 50% to 99%.
10. They state that the resolution can be improved by reducing the scattering strength of the grating coupler. This only improve the longitudinal resolution. The transverse resolution can only be improved by using a wider grating coupler. However, a wider and long grating coupler (large volume) means the scanning range will be smaller and the response be slower. The authors may comment on this trade-off.
11. Fig. S7(b) is not clear. It's hard to find the two primary arms in the figure. It is better to show the entire L-shaped cantilever. Alternatively, they may also show the temperature profile along a planar cut plane.

RESPONSE TO REVIEWER COMMENTS

We thank the reviewers for their comments and we appreciate their support for the device concept. We sincerely apologize for the delay in submitting our response. Please find our responses below in blue.

Reviewer #1 (Remarks to the Author):

...

The work and its conclusions appear well supported by the data. I agree that this is original research towards visible light beam scanning with large field of view and low power consumption. And this work will be of significance to enrich the development of power-efficiency solutions in visible beam steering applications in AR, microscopy and light detection and ranging (LiDAR) etc.

However, the impact and significance may not meet the standard of Nature Communications provided that,

a) the FOV seems limited by undercut of silicon substrate;

While the field of view (FOV) in this set of devices was predominantly limited by the depth of silicon substrate undercut of our current platform, the undercut depth does not invalidate the novelty of the work. We acknowledged and discussed the limitation caused by the undercut depth in the manuscript: Fig. S2c explains that as the tip of the cantilever comes into contact with the substrate, the slope of the curves shown in Fig. 2 (and thus the displacement for a given power) drops. The FOV can be extended to wider angles simply by using a deeper silicon undercut in the future. In simulation, a 1mm long rectilinear cantilever with a deeper silicon undercut of 120 μm would scan 52 degrees under 30 mW applied electrical power. We have included the following sentence in the Discussion section:

Simulations show that with a 120 μm -deep undercut, a 1mm long rectilinear cantilever would achieve a deflection of 52° with 30 mW of applied electrical power.

b) a tradeoff between scanning rate and the steering range, and the scanning rate was limited due to the mechanic nature of the cantilever in the range of only 10-100 kHz;

The trade-off between the scan rate and range applies to all mechanically actuated devices. Indeed, MEMS beam scanners are relatively slow compared to non-mechanical beam steering devices. For instance, a large MEMS-based LIDAR recently reported in Nature [16], is capable of steering the beam only at ~ 100 kHz. This is slower than electro-optical phased arrays which can potentially function at GHz frequencies at infrared (IR) wavelengths.

It is important to keep in mind that our devices operate in the visible spectrum and not IR. The high-speed beam steerers functioning at IR wavelengths typically use tunable lasers and/or phase shifters to steer the beam in two dimensions [23-27]. These functionalities are not yet readily available in the visible spectrum: A rapidly and widely tunable visible laser has not been demonstrated yet, and waveguide phase shifters in photonic integrated circuits for the visible range are still in their infancy. Currently, waveguide phase shifters in the visible range mainly rely on the slow thermo-optic effect [31] or liquid crystals [39], both of which are at least an order of magnitude slower than the resonance frequency of the devices demonstrated in our manuscript. Therefore, the demonstrated scanning rates of the micro-cantilevers in this work are a significant improvement to the state-of-the-art visible light beam scanners [31,44].

Besides the issue that conventional methods of non-mechanical beam steering are not feasible in the visible spectrum, the requirements of laser scanners are different in this wavelength range than IR: 10s of kHz scanning frequency is sufficient for bio-applications and laser scanning displays. Finally, because of

their small volume, the MEMS cantilevers have a higher resonant frequency than most commercially available mirrors (which have resonant frequencies of the order of 1kHz in the ambient). A smaller mirror could reach higher resonant frequencies, but the optical alignment would be more challenging. Please see the answer to Q1 below.

c) also, the size of cantilever basically lies in the micron-meter scale, which is highly susceptible to the ambient variations such as the temperature change. The operation reliability should be verified.

Reliability and sensitivity to the ambient conditions, such as pressure and temperature, are often concerns in MEMS reliability. Operation reliability is best tested in packaged devices, which mitigate some contributors to the device degradation (e.g., contamination, corrosion). As reported in the manuscript, we have studied the operation of the cantilevers in different temperatures (ranging from -263°C to room temperature), in an ultra-high vacuum chamber, and over extended periods of time at their respective resonance frequency. Over about 1 billion cycles at room temperature (3 days, for both L-shaped and 1mm long rectilinear cantilever), we did not observe any degradation in the angular steering range. Furthermore, in our device (as is, unpackaged), the main source for MEMS degradation would likely be the residual stress. The residual stress is more significant at low temperatures (10K) rather than at room or elevated temperatures, as evidenced by the higher initial deflection angle of the cantilever at 10K (Fig. 5(a)) and by the high temperature differential between the cantilever and its surrounding environment. Even under these conditions, we also did not observe any degradation in the steering range after about 7 million switching cycles at 10 K (20 minutes). A more comprehensive test of the operation reliability will be a part of future work.

Regarding to the above, I WOULD NOT recommend to publish this work in Nature Communications unless below comments are addressed:

1)The authors illustrate the circuit of compact footprint, broadband operation and low power consumption. However, I can not see the improvement of such a monolithic design in compared to a combination design of an on-chip laser with an external MEMS reflector. However, the cost and fabrication complexity are distinct.

The main benefits between the monolithic design and an external reflector are (1) flexibility to co-integrate with PICs to achieve functionalities, such as wavelength multiplexing and modulation, without assembly; (2) reduced packaging complexity, due to point (1) and especially when the lasers are integrated into the photonic integrated circuit (PIC); (3) compatibility with wafer-scale manufacturing for both the MEMS, PICs, and their co-integration; (4) generally a higher resonance frequency due to the smaller volume of the monolithically integrated MEMS.

To clarify these points, we have added to the Discussion section:

In comparison to discrete MEMS scanning mirrors co-packaged with a laser diode, the advantages of the monolithically integrated MEMS PIC approach are (1) flexibility to co-integrate PICs to achieve passive and active functionalities, such as wavelength multiplexing, filtering, modulation, and detection, into a single chip without the need to assemble micro-components (e.g., dichroics, lenses); (2) reduced packaging complexity, especially when the lasers are integrated on the chip (a recent demonstration in the near infrared range near 980 nm is reported in (Tran et al. 2022), and we are also developing heterogeneous laser integration approaches for our platform); (3) compatibility with wafer-scale manufacturing for the MEMS, PIC, and their co-integration, and (4) generally a higher resonance frequency due to the smaller volume of the

cantilever compared to a mirror (smaller mirrors are possible but require tighter alignment tolerances).

2) I would recommend the authors to provide a group of data to show the reliability of microcantilevers at different temperature ranges, such as the beam deflection without power input at 20 - 70 °C?

We agree that systematic reliability and accelerated aging tests can be undertaken in future work (at different temperature and humidity levels, as well as mechanical shocks). For a preliminary reliability test, we continuously operated the cantilevers driven on resonance for three days, equivalent to about 1 billion cycles in the ambient environment and about 20 minutes (7 million cycles) in a low temperature environment of 10 K. The cryogenic reliability is expected to be significantly worse than high temperature since the vacuum reduces the thermal conductivity of the device and the internal stresses are much higher, yet no degradation was observed over 20 minutes.

Fig. R1a shows the calculated initial deflection angle of the cantilever at temperatures between 20 and 80°C. Due to residual film stresses, the cantilever beam is bent up upon release at room temperature, and this deflection angle and film stress increases as the temperature is reduced. Heating reduces the stress and the beam bends downwards.

Importantly, the maximum stress in the device (which occurs at the clamping point) is critical for the device reliability, as an increased shear stress can result in cracks or fracture, as well as fatigue after billions of cycles. To quantify the stress as a source of unreliability, using COMSOL Multiphysics, we calculated the von Mises stress along the device (Fig. R1b) at different temperatures between -263.15 to 80 °C. Although the dominant stress component causing breakage is the Cauchy stress normal to the cantilever, we can take the von Mises stress as a better measure of device longevity since it takes other minor sources of stress into account. The results are shown in Fig. R1c at different temperatures. The maximum stress in the clamp is almost an order of magnitude higher at 10 K, compared to the room temperature. This is due to the upward bending of the cantilever, which results in an increased stress on the fixed point. On the contrary, by increasing the temperature, due to the reduction of the cantilever deflection, the stress is drastically reduced for ~2 orders of magnitude at 80 °C compared to the room temperature. Therefore, we believe that increasing the temperature should not negatively affect the reliability of the device.

Apart from stress, the bimorph reliability also depends on various failure mechanisms including fatigue and creep. Creep may cause long-term deformation if the material is subjected to heating for a long time. However, as a general rule, creep deformation starts to matter at temperatures greater than 35% of the melting point of material [R1, R2], which is beyond the performance of our heaters. Also, fatigue is increased at higher temperature and higher stress levels [R3, R4]. Nevertheless, we agree that a systematic reliability test in a more mature and packaged device in the future would provide useful information on the limits of the device.

Fig. R1: **a** Simulated initial angle of the L-shaped cantilever at $T=20$ to 80°C . **b** Calculated von Mises stress (in Pascal) along the device at 10 K (top) and room temperature (bottom). The inset shows the clamped point with the highest stress. **c** Temperature dependency of the von Mises stress at clamp.

To clarify this, we have added in the Discussion section of the manuscript:

The reliability of the cantilevers and other MEMS structures implemented in the PIC platform can be systematically characterized in the future in more mature and packaged devices as a function of temperature, humidity levels, mechanical shocks, applied electrical power, and the scan range and speed [48]. In the ideal operation scenario, the microcantilever tip should not come into contact with the substrate, which can be achieved in the future with a deeper undercut. Due to the difference in the coefficients of thermal expansion of SiO_2 and Al, the cryogenic conditions in our experiments (at 10 K) increase the intrinsic stress in the cantilevers compared to an elevated temperature (in simulation, the stress at clamping is ~ 10 times higher at 10 K compared to the

room temperature). Although not a comprehensive reliability characterization, our device measurements at 10 K for 20 minutes and in the ambient for 3 days did not exhibit any observable degradation; these observations are a promising indication of the robustness of the microcantilever-integrated photonic circuits.

3)The circuit contains a single grating coupler of 10 μm wide and 25 μm long. This single grating element will be fundamentally diverged in far field, and the beam divergence is noticeable in free space, particularly in transverse direction, highly dependent on the aperture size. Without any beam concentration on the emitting element, just like OPA, this will be a big issue for practical application. Even though the smaller divergence angle with longer grating is claimed by the authors, there will be additional issue appearing together with longer gratings. 0.01° divergence is delivered by 8 mm grating aperture reported in [C. V. Poulton, et al., IEEE J. Sel. Top. Quantum Electron 28(5), 6100508(2022).] The authors should seriously discuss how this millimeter size grating is bended by a much bigger microcantilever. Will the thermal actuation affect the optical properties of millimeter-size grating? How should the thermal isolation be conducted? What scanning rate can be expected on this bigger cantilever?

We agree with the comment. As mentioned by the reviewer, reaching a low divergence of 0.01° is challenging using our proposed structures. Apart from arising thermal isolation challenges, based on our model, an eight-millimeter-long rectilinear cantilever will have a small actuation speed of ~ 100 Hz. Nevertheless, as mentioned in the text and shown in Section 5 of the Supplementary document, elongating the grating coupler only to 100 μm along the demonstrated cantilevers can reduce the divergence angle in longitudinal direction to 0.2°. Moreover, unlike in a LiDAR, a beam size as large as 0.2° can be sufficient for some visible light applications such as photostimulation in (neuro)biology. To reduce the beam divergence, either a larger aperture (grating) along with a different actuator design is employed, or an external optic can be used to collimate the beam.

4)Can the authors explain more about how you assumed the initial strain of cantilevers to match the experimental observation by seeking the deposition temperature of Al in the simulations?

The initial strain of the cantilevers at room temperature, is mainly the result of a) the difference between the expansion coefficients of Al and SiO₂, and b) Al having been deposited at a temperature other than room temperature. COMSOL Multiphysics allows for defining a “reference temperature” in the MEMS module, that is, the temperature at which there is no thermal strain in the solid (Fig. R2).

Fig. R2: Reference temperature allowing for modeling the initial strain of the MEMS.

By sweeping this model input (T_{fab}) we could estimate the temperature at which metal was deposited. At T_{fab} close to 80°C , the initial strain of the cantilever becomes equal to what we had experimentally observed using the Fourier measurement setup, in the absence of an applied voltage.

5) Please re-check the caption of Figures 2b and 2c upon the illustration of primary and secondary arms. The illustration should be consistent with Figure 1d.

Thank you for spotting this typo in the caption. The caption is now corrected with “ θ ” with “ φ ” switched.

6) In Figure 3b, I suppose that all 4 samples were prepared under same conditions. Then, the initial strain (right patterns) of GC tips should be of slight increase regrading to the growth of arm length. This can be verified in Figure 3b with $L_{dev} = 300, 500$ and $800 \mu\text{m}$. However, the initial value for $L_{dev}=1000 \mu\text{m}$ seems not greater than others. Can you explain why?

We appreciate the detailed observation of the reviewer. There is a subtle point regarding the Fourier image of the output of the devices shown in Fig. 3b specifically about the $1000 \mu\text{m}$ long cantilever. In Fig. 3b, we aimed to demonstrate the change in the angular deflection, $\Delta\theta$, (as plotted in Fig. 3c) and not the absolute angle. Only in the case of the cantilever with $L_{dev} = 1000 \mu\text{m}$, $\Delta\theta$ was so large that the right spot was nearly outside of the numerical aperture of our measurement setup. Therefore, we slightly shifted the Fourier imager to capture the full span of the grating coupler far field in the center, this also helps to get a more precise calculation of the $\Delta\theta$. This has caused the slight left-shift spotted by the reviewer. We have added the following to the caption:

For the $L_{dev} = 1000 \mu\text{m}$, the measurement setup was slightly shifted to capture both spots.

7) The dynamic motion of cantilevers must be distinct from their static response. If possible, please provide the frequency-amplitude(strain) response of the rectilinear cantilever tests at one certain arm length. And also, for $L_{dev}=1000 \mu\text{m}$, please also clarify why the resonance strain is less than static strain, if you compare the results in Fig. 3f and Fig. 3c.

Yes, to report both the static and dynamic responses in the manuscript, we have separately simulated, measured and reported the “DC response”, as well as the “time domain response” in the form of rise/fall time. Additionally, we have measured and reported the devices performance at their resonance frequencies. In this case, an extensive frequency response measurement seemed slightly redundant. Nevertheless, in order to address the comment of the reviewer, here, we present the measured frequency dependent output ($\Delta\theta$) of the rectilinear cantilevers with two different lengths ($L_{dev} = 500 \mu\text{m}$ and $1000 \mu\text{m}$). In both cases, a continuous sinusoidal signal with $2 V_{pp}$ amplitude and an additional 1 V offset is applied to the device under test, and the swing in the output is measured using the Fourier measurement setup shown in Fig. 3a.

The results are shown in Fig. R3. The -3dB bandwidth of the $500 \mu\text{m}$ and $1000 \mu\text{m}$ devices are 160 Hz and 300 Hz , respectively. It is important to mention that apart from the device length, the main limiting factor of the device speed is the thermal time constant, which can be mitigated provided that the cantilevers actuate using electro-static mechanism.

Fig. R3: Measured frequency response of the 500 μm and 1000 μm long rectilinear cantilevers with 2 Vpp applied sinusoidal signal.

Regarding the reviewer's other question on why the measured dynamic strains of the cantilevers (especially in the case of $L_{\text{dev}} = 1000 \mu\text{m}$) is smaller than their static strains, one should consider that at high input frequencies (e.g., at resonance frequency), the far-field pattern at 0 mW, is not equal to the case of static 0 mW input power: Under a constant 0 mW applied power (static measurement), the cantilever has enough time to reach to the room temperature, and thus, deflection reaches its maximum value. In contrast, in the dynamic measurements, even at 0 mW applied electrical power, the cantilever cannot reach its maximum value (since the cantilever temperature is still higher than room temperature) and thus the total dynamic displacement becomes smaller. For the dynamic measurements at the resonance frequencies, we limited the input power of the cantilever so that its tip does not contact the Si substrate to avoid stress relaxation which might damage the cantilever at high modulation rates. The reduction of the dynamic strain is also observed for the L-shaped device (comparing the far-field patterns in Fig. 4c with Fig. 4g). The DC-component of the applied electrical power at resonance does not contribute to $\Delta\theta$, but adds a constant offset to the downward deflection to the cantilever, which reduces the overall dynamic range of the scan.

8) Would you please roughly explain the possibility of cantilever damage under large strain resonance, since the cantilever may have chance to knock onto the silicon surface at high frequencies. This test would support the robustness of the microcantilevers.

We agree that contacting the Si substrate at high speeds can result in a large stress relaxation over time, which in turn affects the device performance in the long run and compromises the device longevity. To avoid this, in this work (as explained under comment 7 of the first reviewer), we avoided applying high electrical power at resonance frequency at the cost of reducing the dynamic strain. All the measurements on resonance were limited to the dynamic strain shown in Fig. 3f. Nevertheless, we would like to emphasize the main limiting factor in this case was the Si undercut depth in this current platform. In future designs, the undercut will be sufficiently deep such that the cantilever tip will not come into contact with the Si substrate. The reliability of the device and the damage threshold of the design should be characterized in the nominal operating conditions, without touching the Si substrate.

9) Frequency response of cantilevers under cryogenic conditions should be provided, such as the response time as well as the resonance steering range.

Although it would have been interesting to repeat the same measurements performed to address comment 7 of the first reviewer in cryo-chamber, they are practically challenging to be performed at cryogenic temperatures. For the reasons mentioned in Section 6 of the Supplementary information, the cooling process needed to be done at a low rate. Therefore, only a small amount of liquid helium remained to keep the device stably at a low temperature for a full-scale frequency domain measurement. Nevertheless, as stated in the manuscript, we measured the first resonance frequency of the L-shaped cantilever to be 16.76 kHz, as well as the steering range which is shown in Fig. 5(b) and (c). The main motivation of running the device at low temperatures was to test the device under an increased level of stress. Duplicating all the room-temperature measurement in cryogenic temperatures was unfortunately not possible with our measurement setup and conditions.

[R1] Basu, S., & Debnath, A. K. (2019). Advanced ultrasupercritical thermal power plant and associated auxiliaries. *Power plant instrumentation and control handbook*, Elsevier, 893-988.

[R2] Ashby, M. F., Shercliff, H., & Cebon, D. (2018). *Materials: engineering, science, processing and design*. Butterworth-Heinemann.

[R3] Wang, P., Liu, Y., Wang, D., Liu, H., Liu, W., & Xie, H. (2019). Stability study of an electrothermally-actuated MEMS mirror with Al/SiO₂ bimorphs. *Micromachines*, 10(10), 693.

[R4] Kahl, S., Ekström, H. E., & Mendoza, J. (2014). Tensile, fatigue, and creep properties of aluminum heat exchanger tube alloys for temperatures from 293 K to 573 K (20 C to 300 C). *Metallurgical and Materials Transactions A*, 45(2), 663-681.

Reviewer #2 (Remarks to the Author):

The authors demonstrated a laser beam scanner by using microcantilevers embedded with silicon nitride nanophotonic circuitry. They achieved broadband, one- and two-dimensional steering of light with a broad visible wavelength range. The device has a compact size and low power consumption. It was realized in an active photonic platform on 200-mm silicon wafers. The design is novel and interesting. Good performances were obtained. I have the following questions and comments.

1. It is not clear how the electrical connections were formed in the rectilinear cantilever and L-shaped cantilever. In order to generate heat by the applied voltage, the two ends of the TiN heater must be connected the aluminum electrodes. Is the connection through the via in the cantilever? In Fig. S3, they provide the circuit diagrams for the L-shaped cantilever, but it is hard to match the circuit diagrams with the structure diagram.

A. Rectilinear cantilevers:

To further clarify the connections, we show the simplified layout of the 1 mm long rectilinear cantilever in Fig. R4. As mentioned by the reviewer, the voltage is applied to both ends of the TiN heater. The GND contact pad connects to TiN using via (Via2)). The Vapp pad connects to the far side of the TiN heater in the cantilever using M2 (that is the 2 μm thick Al on top of the cantilever) and Via2. The electrical signal is sent to the far side of the cantilever via M2.

Fig. R4: Layout of the 1 mm long rectilinear cantilever.

B. L-shaped cantilever:

The layout of the L-shaped device is shown in Fig. R5 and consists of three heater sections: Heater1 and Heater2 are placed respectively in the left and the right primary arms, and Heater3 in the secondary arm.

Ideally, one should be able to individually address each heater independent of the other two heaters. We designed the layout to achieve this goal by defining a via at the far end of Heater1. In this way, the voltage reaching Heater1 is equal to $(V2 - V1)$, as the near end of Heater1 is directly connected to $V1$. Similarly, the voltage drop on Heater2 is $(V4 - V3)$. The far end of Heater2 is connected to $V3$ using a Via (top-left inset in Fig. R5) and Al layer, and its near end is directly connected to $V4$.

The voltage drop along Heater3 is determined by the difference between $V2$ and $V3$: The near end of Heater3 is connected (using via and low resistivity Al) to $V3$, while the far end of the Heater3 is connected (using a Via, and low resistivity Al on top of secondary and primary arms) to $V2$.

We have assumed that the voltage drop in the Al is negligible compared to the voltage drop in the TiN layer. This is a valid assumption, since the resistivity of the Al layer (due to its $2 \mu\text{m}$ thickness compared to 150 nm thickness of TiN) is ~ 2 orders of magnitude smaller than the resistivity of TiN layer.

Fig. R5: Layout of the L-shaped cantilever. The placement of the vias allows for independent addressing of the individual heaters inside the device.

2. In Fig. 1(a), the cantilever part includes a part of the silicon substrate (the undercut is not just under the SiO₂-Si interface). How can this be realized in the device fabrication?

We apologize for the original figure, which contained this mistake. Below is the corrected cross-section, which is the current Fig. 1(a).

3. For the L-shaped cantilever, why did they design two primary arms instead of one? What is the lateral separation distance between the two primary arms? How will this distance affect the beam scanning in both longitudinal and transverse directions? When electrical power is applied to the primary arms to get beam steering along the φ -axis, is the power evenly applied to the two arms?

Our initial idea was also to realize 2D beam steering using a simpler L-shaped cantilever, with only one primary arm, and one secondary arm. The problem was that in that simpler configuration, we could not come up with any possible circuitry (using only two layers: Al and TiN) which allows for independently addressing the two heaters. In this simpler configuration, the Al line on the primary arm and the secondary arm is shared, and therefore has the same voltage. The question was “how to apply two different voltages to the near-end and far-end of the secondary arm, with only one Al line?”. One method to solve this issue could have been to increase the width of the primary arm, so that two different Al lines can be placed atop. Our simulations showed that in this case the angular efficiency of the primary arm ($\Delta\varphi / V$) drops. This is mainly due to the fact that the heater in the primary arm had to heat up a larger volume, while the distance between Al lines naturally resulted in a smaller effective area (as the central part of the oxide in the primary arm was not covered with Al). Moreover, increasing the width of the suspended arms is not generally ideal, since it increases the risk of Si not being fully undercut. Therefore, we decided to split the primary arm into two, so that each set of arms can be independently addressed electrically.

In our device, the gap between the two primary arms was chosen to be 40 μm . Ideally, the distance between the arms should be as small as possible. This is due to the fact that increasing the gap between these two arms in practice results in reduction of the effective length of the secondary arm, and thus reduces the maximum achievable $\Delta\theta$. The reason why we did not choose a gap smaller than 40 μm was that we wanted to be sure that both arms would have a full Si undercut. In our measurements, we applied the power to the right-side primary arm. However, the simulations (partially presented in Section 3 of Supplementary material) showed that applying the electrical power to each one of the primary arms could have roughly the same effect regarding the achievable $\Delta\varphi$.

4. According to Fig. 1(d), the SiN waveguide is first routed through the left primary arm and then the secondary arm before terminated at the grating coupler. In other words, there is no SiN waveguide in the

right primary arm. The two primary arms are not identical. Will this asymmetry make the tuning efficiency different for the two primary arms and hence cause θ -axis tilt under ϕ -axis tuning?

Apart from thermal crosstalk, as mentioned by the reviewer, the asymmetry between the two primary arms can also result in a non-ideal tilt of the grating coupler and thus the output beam. We agree with the reviewer's opinion that the asymmetry between the two arms due to the SiN waveguide is partially responsible for the non-ideal strain of the L-shaped cantilever. This negative effect should have been avoided by placing a dummy SiN waveguide inside the right-hand-side primary arm, identical to the real waveguide. Nevertheless, since the SiN waveguide occupies only a small fraction of the total volume of the primary arm ($\sim 0.06\%$), we believe that this effect is less pronounced than the unavoidable thermal crosstalk between the two sets of the arms.

5. In Fig. 4(c), the applied electrical voltage signals are two sinusoidal waves. The voltage expressions in the caption are wrong (the sin function missing).

Thank you for pointing out the typo. We have fixed the equations to the following:

$$V_{\theta} = 2[\sin(2\pi f_1 t) + 1] \text{ V} \quad V_{\phi} = 1.5[\sin(2\pi f_2 t) + 1] \text{ V}$$

6. What is the purpose of characterizing the beam scanner in cryogenic conditions? Is there any special application that requires the beam scanner to work in a very low temperature?

We characterized the device in cryogenic conditions since it was unclear if the electrothermal actuation would be effective at low temperature (the explanation is in Section 6 of the Supplementary Information), and quantum applications (i.e., the manipulation of atoms and ions) may require the photonic circuit to be inserted into the cryostat. We have clarified in the main manuscript by revising the paragraph describing the cryogenic experiments:

Finally, we characterized our beam scanner in cryogenic conditions for several reasons: (1) the low temperature increases the intrinsic stress in the cantilever and poses a higher risk for device failure, (2) electrothermal devices may not work in a cryostat since their thermal budget can overwhelm the cooling power of the cryostat, and (3) some applications, for example in quantum control, may prefer to have the PIC inside the cryostat (Niffenegger et al. 2020; Mehta et al. 2020).

7. In Fig. S2(b), the temperature suddenly drops at around 470-micron distance along the cantilever. According to my understanding, this end is not clamped. Then what is the reason that causes this temperature drop?

The hottest points inside the suspended arm are the ones which are closest to the TiN heater itself. As can be seen in Fig. R4 (right inset), we did not extend the TiN heater to the tip of the cantilever, but it stops $\sim 32 \mu\text{m}$ shy of the cantilever tip. Therefore, the temperature progressively drops in the distal zone of the cantilever end.

The reason that the TiN layer is not extended to cover the entire length of the cantilever is that TiN (same as Al) is not transparent for the visible light, and thus should not cover the grating coupler, where the output beam is emitted out.

8. Fig. S3(d) caption says the graph shows the calculated ϕ -axis angular tilt of the grating coupler. However, the y-axis of the graph is labeled as $\Delta\theta$ (it should be $\Delta\phi$).

Thank you for pointing out the error. The caption in Fig. S3(d) has been corrected.

9. If the duty-cycle of the applied square pulses is varied between 1% to 50%, the rise time can be measured, but how can they get the fall time? According to my understanding, in the fall time measurement, the duty-cycle should be varied between 50% to 99%.

This is correct. We have varied the square pulse between 1% to 50%, and then from 50% to 99% in order to measure respectively the rise time and the fall time of the devices. We corrected the text in Supplementary Section 4 accordingly:

In these measurements, the applied square pulse had a period of 20 ms and the duty cycle was varied between 1% to 50% to measure the rise time, and 50% to 99% to measure the fall time.

10. They state that the resolution can be improved by reducing the scattering strength of the grating coupler. This only improve the longitudinal resolution. The transverse resolution can only be improved by using a wider grating coupler. However, a wider and long grating coupler (large volume) means the scanning range will be smaller and the response be slower. The authors may comment on this trade-off.

This is a neat idea to increase the resolution of the output beam, which is not mentioned in the manuscript. Based on the simulations, solely increasing the width of the cantilever does not immediately affect the resonance frequency of the device. This can also be verified using the equation for calculating the first resonance of the single clamped cantilever [R5]

$$f_0 \approx 0.162 \frac{d}{L} \sqrt{\frac{Y}{\rho}},$$

where d is the thickness, L is the length, Y is the young's modulus and ρ is the density of the cantilever. The first resonance frequency can be approximated to be independent of the cantilever width. Nevertheless, as mentioned by the reviewer, increasing the width linearly decreases the angular efficiency of the device (in terms of $\Delta\theta/1\text{mW}$) due to an increased volume of the cantilever. This means if a 20 μm wide cantilever requires 10 mW electrical power to yield beam steering of 15 degrees, ~ 20 mW electrical power will be required to achieve the same tilt with a 40 μm wide cantilever. Another issue arising from widening the cantilever is that it becomes increasingly challenging to ensure a full Si undercut under the suspended beam. This problem might be addressed to some extent by introducing holes inside the wide beam. Nonetheless, it will further complicate the design and might compromise the longevity of the device. Based on these issues, we decided against improving the transverse resolution by widening the suspended arms. Such designs can be explored in future wafer runs.

11. Fig. S7(b) is not clear. It's hard to find the two primary arms in the figure. It is better to show the entire L-shaped cantilever. Alternatively, they may also show the temperature profile along a planar cut plane.

Thank you very much for your observation. We accordingly modified Fig. S7(b) and included the temperature profile in (c).

[R5] Stenne, G. (1991). Resonant silicon sensors. *Journal of Micromechanics and Microengineering*, 1(2), 113.

REVIEWER COMMENTS

Reviewer #1 (Remarks to the Author):

The authors have addressed most issues in this response. But there are still two main concerns to figure out, as below:

1. When it is compared to traditional MEMS mirror, it is not suitable to pay too much attention to integration of MEMS PIC. The question is what difference of traditional mirror versus grating cantilever in this work. In fact, the radiation efficiency of waveguide grating is fundamentally slower than mirror. The waveguide grating normally have low radiation efficiency of <50%, however, the reflection efficiency of mirror is usually >90%. When this scanner used in AR display or photostimulation in biosensor, the efficiency is critical for display brightness or SNR for biophotonics respectively.

2. About the bending performance of long grating cantilever, the authors didn't provide enough evidence for scanning properties of mm-size grating. It is mentioned that 100um-long cantilevers can provide divergence angle of 0.2 degree. However, this is far below the practical requirements for AR display. The basic requirements are 60 pixels per degree as mentioned in [Xiong, J., Hsiang, EL., He, Z. et al. Augmented reality and virtual reality displays: emerging technologies and future perspectives. Light Sci Appl 10, 216 (2021)]. To meet this resolution, the cantilever need be extended to mm-size scale, but the actuation speed will be slow as ~100 Hz. This slow speed cannot provide enough frame rate for AR display. Also, the authors mentioned external optics are used to collimate the beam. But how to design a dynamic optics to keep following the bending cantilever need to be figured out in details. Typical lens set is nearly impossible to capture the fluctuating focus point of bending cantilever.

I will not recommend this manuscript to publish in current form, till these fundamental issues are addressed properly.

Reviewer #2 (Remarks to the Author):

The authors have responded to the questions in my previous peer review report. The manuscript has been revised and more information has been added. I recommend it for publication in Nature Communications.

RESPONSE TO REVIEWER COMMENTS

We thank the reviewers for their comments and we appreciate their support of our work. Please find our responses to the remaining questions of Reviewer #1 below in blue.

Reviewer #1 (Remarks to the Author):

1. When it is compared to traditional MEMS mirror, it is not suitable to pay too much attention to integration of MEMS PIC. The question is what difference of traditional mirror versus grating cantilever in this work. In fact, the radiation efficiency of waveguide grating is fundamentally slower than mirror. The waveguide grating normally have low radiation efficiency of <50%, however, the reflection efficiency of mirror is usually >90%. When this scanner used in AR display or photostimulation in biosensor, the efficiency is critical for display brightness or SNR for biophotonics respectively.

Thank you for clarifying the purpose of the question. We agree with the reviewer that the radiation efficiency of the grating couplers is lower than the traditional MEMS mirrors unless there are extra features like back reflectors. Based on this comment, to emphasize this advantage of MEMS mirrors over integrated emitting devices, we added the following sentence to the manuscript:

On the other hand, in general, MEMS mirrors have a higher radiation efficiency compared to grating couplers (typically ~90% vs. ~50%, respectively) unless extra features, such as back reflectors, are incorporated on chip, and mirrors have broadband reflection. Thus, the choice between using a co-packaged MEMS mirror and a fully integrated solution depends on the specific application.

Our approach indeed cannot replace traditional MEMS mirrors in their entirety. Based on their advantages and drawbacks, one might decide to use either solution for a specific application.

All being said, it is worth mentioning that despite the inferior transmission efficiency of on-chip emitters, integrated solutions for MEMS beam scanners have become a major interest of the community (e.g. [16] published in Nature last year). There are several technical reasons (apart from integration of MEMS with PIC) for such a push towards a fully integrated solution, which we briefly explained in the paper:

... (2) reduced packaging complexity, especially when the lasers are integrated on the chip (a recent demonstration in the near infrared range near 980 nm is reported in [46], and we are also developing heterogeneous laser integration approaches for our platform); (3) compatibility with wafer-scale manufacturing for the MEMS, PIC, and their co-integration, and (4) generally a higher resonance frequency due to the smaller volume of the cantilever compared to a mirror (smaller mirrors are possible but require tighter alignment tolerances).

Apart from the aforementioned advantages, our proposed devices have a significantly smaller footprint compared to MEMS mirrors. The size of the largest rectilinear cantilever shown in this work is ~0.05 x 1.0 mm², which is smaller than the typical footprint of MEMS mirrors (> 1 mm x 1 mm). The compactness becomes increasingly important in the applications where a large number of beam scanners are required to function simultaneously.

2. About the bending performance of long grating cantilever, the authors didn't provide enough evidence for scanning properties of mm-size grating. It is mentioned that 100 μ m-long cantilevers can provide divergence angle of 0.2 degree. However, this is far below the practical requirements for AR display. The basic requirements are 60 pixels per degree as mentioned in [Xiong, J., Hsiang, EL., He, Z. et al. Augmented reality and virtual reality displays: emerging technologies and future perspectives. Light Sci Appl 10, 216 (2021)]. To meet this resolution, the cantilever need be extended to mm-size scale, but the actuation speed will be slow as \sim 100 Hz. This slow speed cannot provide enough frame rate for AR display. Also, the authors mentioned external optics are used to collimate the beam. But how to design a dynamic optics to keep following the bending cantilever need to be figured out in details. Typical lens set is nearly impossible to capture the fluctuating focus point of bending cantilever.

We agree with the assessment of the reviewer on the minimum required aperture size of the emitter in order to achieve a higher resolution. Based on our simulation results shown in Fig. S6a, a cantilever length of at least 1 mm is required to reach an acceptable resolution (with a divergence angle less than 0.03 degree) provided that a low scattering grating coupler is extended along the cantilever. The reviewer correctly points out that such a long cantilever (as we have also reported in the paper) would have a large time response (as we have measured 5.72 ms rise time) and thus would be too slow for some applications e.g., AR displays, where at least an order of magnitude faster response is required.

One way to overcome the low speed of the long cantilevers is to overdrive the device. This method has been previously used to get a faster time-response from thermal phase tuners for instance in [R1]. To show the feasibility of using this solution to improve the time response of the cantilevers, we have simulated the fall time of the 1 mm long rectilinear cantilever, under the applied voltage shown in Fig. R1(a). The resulting output angle of the output beam is shown in Fig. R1(b). The fall time is reduced to 0.5 ms, which is about one order of magnitude reduction compared to the normal fall time of this cantilever. To improve the rise time, in the absence of a thermoelectric cooler (TEC), large metal patches should be used on the clamped side of the cantilever for better heat sinking.

Fig. R1: **Simulated fall-time with overdriving.** **a** Applied electrical power. The dashed red curve shows the applied power with an overdrive, which is 90 mW before $t = 0.3$ ms. The blue curve shows a simple step function. **b** Simulated output beam angle, as a result of two types of input power signals. Overdriving the applied power shortens the fall time to 0.5 ms.

Another method to reduce the lateral divergence of the output beam, is to use an array of grating couplers to effectively form an optical phased array. The lateral steering by the phase difference can be achieved

by slightly displacing the grating coupler vertically. A final method to raster an image is to replace the grating couplers on the cantilevers with edge couplers akin to fiber scanners.

To briefly introduce these concepts for the next steps in the research, we have included the explanation and calculation of overdriving the cantilever (Fig. R1) in the Supplementary Information (Section 5 and Figure S7), and we have also revised the third last paragraph of the manuscript to the following:

The cantilever beam scanners can be extended in several ways... To increase the number of resolvable spots, ... the divergence angle of the output beam along the propagation direction could be significantly decreased by $>10\times$ using weaker and longer gratings (Supplementary Section 5), and the divergence in the lateral direction can be reduced with an array of cantilevers that effectively forms an OPA.... Beyond the demonstrated geometry... other types of light emitters, such as edge couplers (akin to fiber scanners)... can be used.

Indeed, this work opens many opportunities for further device explorations and PIC architectures. We hope our replies have provided sufficient information on the potential next steps, some of which are currently under investigation in our research group.

References

[R1] Geis, M. W., Spector, S. J., Williamson, R. C., & Lyszczarz, T. M. Submicrosecond submilliwatt silicon-on-insulator thermo-optic switch. *IEEE Photonics Technology Letters*, 16(11), 2514-2516, 2004.

REVIEWERS' COMMENTS

Reviewer #1 (Remarks to the Author):

The authors have responded to my concerns. The manuscript has been modified. It can be accepted for publication in Nature Communications.